# Deep learning image analysis for continuous single-cell imaging of dynamic processes in *Plasmodium falciparum*-infected erythrocytes

Sophia M. Frangos [1], Sebastian Damrich[2,5], Daniele Gueiber[1,3], Cecilia P. Sanchez[1], Philipp Wiedemann[3], Ulrich S. Schwarz[4], Fred A. Hamprecht[2] & Michael Lanzer [1] ✉

Continuous high-resolution imaging of the disease-mediating blood stages of the human malaria parasite *Plasmodium falciparum* faces challenges due to photosensitivity, small parasite size, and the anisotropy and large refractive index of host erythrocytes. Previous studies often relied on snapshot galleries from multiple cells, limiting the investigation of dynamic cellular processes. We present a workflow enabling continuous, single-cell monitoring of live parasites throughout the 48-hour intraerythrocytic life cycle with high spatial and temporal resolution. This approach integrates label-free, three-dimensional differential interference contrast and fluorescence imaging using an Airyscan microscope, automated cell segmentation through pre-trained deep-learning algorithms, and 3D rendering for visualization and time-resolved analyses. As a proof of concept, we applied this workflow to study knob-associated histidine-rich protein (KAHRP) export into the erythrocyte compartment and its clustering beneath the plasma membrane. Our methodology opens avenues for in-depth exploration of dynamic cellular processes in malaria parasites, providing a valuable tool for further investigations.

Malaria, caused by the protozoan parasite *Plasmodium falciparum*, remains a major global health challenge affecting hundreds of millions of people worldwide[1]. The dire malaria situation underscores the need for a better understanding of the complex host-parasite interactions leading to disease manifestation in order to develop improved intervention strategies that, e.g., overcome the current drug resistance challenge. In this context, the ability to observe and analyze host-parasite interactions at the single-cell level over time would be a powerful tool[2–4]. For example, time-resolved continuous single-cell imaging would facilitate additional insights into the function of parasite-encoded proteins, particularly if the protein of interest changes its subcellular localization during parasite development or if it assembles into larger complexes over time.

Light-microscopy has been extensively used to investigate cellular processes in *P. falciparum*[5,6]. However, these studies have largely depended on static snapshots, which - even when involving single cell analysis - lack temporal resolution. As a result, information about dynamic processes can only be inferred indirectly in a pseudotemporal manner by aligning images from different cells captured at different time points during the course of a particular biological phenomenon. Although pseudotemporal studies have provided invaluable insights into the biology of *P. falciparum*, such as developmental processes[7] and host cell invasion[8–10], they do not capture real-time dynamics and are, therefore, prone to misinterpretation. Only few studies have thus far attempted to image the parasite over prolonged periods of time in four dimensions[11–13].

The development of long-term single cell imaging for *P. falciparum* is complicated by the intracellular life style and the small size of the parasite. Conventional imaging techniques, which rely on lower-intensity illumination, fail to capture the intricate details necessary for comprehensive analysis[14]. On the other hand, super-resolution microscopic methods require high laser power to achieve the desired resolution[15]. However,

---

[1]Heidelberg University, Medical Faculty, University Hospital Heidelberg, Center for Infectious Diseases, Parasitology, Im Neuenheimer Feld 324, Heidelberg, Germany. [2]Heidelberg University, Interdisciplinary Center for Scientific Computing (IWR), Im Neuenheimer Feld 205, Heidelberg, Germany. [3]University of Applied Sciences Mannheim, Institute of Molecular and Cell Biology, Paul-Wittsack-Strasse 10, Mannheim, Germany. [4]Heidelberg University, BioQuant and Institute for Theoretical Physics, Philosophenweg 19, Heidelberg, Germany. [5]Present address: Hertie Institute for AI in Brain Health, University of Tübingen, Otfried-Müller-Straße 25, Tübingen, Germany. ✉e-mail: michael.lanzer@med.uni-heidelberg.de

prolonged exposure to intense light can induce changes in cell physiology and even result in cell damage[16–18]. *P. falciparum*-infected blood stages are highly light sensitive due to the presence of high amounts of photoactive metabolites, such as heme and riboflavins[19]. To overcome the limitations imposed by the unique biology of the malaria parasite, innovative imaging and image analysis techniques are necessary.

Recent advances in microscope design and data processing now offer new opportunities[20,21]. For instance, next-generation light microscopes operate with reduced light exposure while maintaining high resolution[22,23]. Additionally, machine learning approaches, including deep neural networks, are increasingly used for cell recognition, cell segmentation, and signal tracking[24–26], enabling the automatized analysis of large datasets with high efficiency. These new tools have already been applied in several biological systems, including *Arabidopsis* for development studies[27,28], *Drosophila* for organ development and neural mapping[29], *Caenorhabditis elegans* for gene expression analysis[30], and zebrafish for real-time imaging of organ development[31,32]. In the context of *P. falciparum*, Preißinger et al. have recently demonstrated a neural network capable of identifying individual erythrocytes in multi-cellular two-dimensional images, distinguishing between infected and uninfected red blood cells, and classifying parasite stages into rings, trophozoites, and schizonts[33]. In another study, Geoghegan et al. employed lattice-sheet microcopy, a technique that uses a low dose of light, to investigate parasitophorous vacuolar formation (a compartment separating the parasite from the erythrocyte cytoplasm), capturing this process in a space and time-resolved manner during invasion[13].

In this study, we present a method that uses deep learning on label-free differential interference contrast (DIC) images obtained with an Airyscan microscope to segment erythrocytes and the different asexual blood stages of *P. falciparum*. This approach enabled automatic image analysis and the extraction of three-dimensional spatial and temporal information with high accuracy. It further allowed us to track individual parasites over the entire intraerythrocytic cycle of 48 h. As proof of principle, we monitored the trafficking and sorting of the knob-associated histidine-rich protein (KAHRP) and its assembly into knobs underneath the erythrocyte plasma membrane over the entire asexual intraerythrocytic cycle.

Knobs play a pivotal role in the pathophysiology of *P. falciparum*, by serving as a platform for anchoring adhesins of the *P. falciparum* erythrocyte membrane protein family 1 (PfEMP1) to the actin-spectrin membrane skeleton of the host erythrocyte[34,35]. Knobs appear around 20 h post invasion[34]. Hours later, thousands of them are present on the erythrocyte surface[36], mainly at positions where actin protofilaments have been removed from the host cytoskeleton[37]. As a consequence, infected erythrocytes cytoadhere to the endothelial lining, uninfected erythrocytes, and platelets[34]. By avoiding passage through the spleen, they escape splenic clearance mechanisms, but on the other hand, cause severe pathology to the patient, such as localized hypoxia in occlude capillaries or the syndromes associated with maternal or cerebral malaria[38]. The dynamics of KAHRP export, localization, and assembly are not understood well and monitoring the process in single cells throughout the replicative cycle opens the perspective of better understanding and possibly even intercepting this central process in the pathogenesis of malaria.

## Results

### A neural network for segmentation of *P. falciparum* infected erythrocytes

Figure 1 depicts the experimental workflow for monitoring and analyzing a dynamic process in *P. falciparum*-infected erythrocytes at the single cell level and throughout the entire asexual intraerythrocytic developmental cycle, as exemplified by the export of KAHRP and its localization in the erythrocyte compartment. This workflow encompasses the following steps: i) acquisition of 3D-stacks of single cell images using an Airyscan microscope alternating between the DIC mode and the fluorescence mode (Fig. 1A); ii) automatic cell segmentation utilizing a neural network based on Cellpose (Fig.1B); iii) analyzing the spatial and temporal dynamics of the process in four-dimensions throughout the 48 h long replicative cycle (Fig.1C and D); and iv) 3D rendering of the captured images for visualization and analysis (Fig.1C).

The acquisition of 3D-images for many single cells throughout the intraerythrocytic cycle creates datasets that are too large for manual analysis. We, therefore, decided to use a neural network to segment *P. falciparum* infected erythrocytes and delineate the erythrocyte plasma membrane, the erythrocyte cytosol, and the parasite compartment. Our choice fell on Cellpose. Cellpose is a convolutional neural network (CNN) designed for cell segmentation tasks and pretrained on a diverse set of biological images. Accordingly, only a few annotated examples were needed to re-train it to our use case. Moreover, Cellpose can analyze both 2D and 3D images, making it a perfect tool for our 3D transmitted light images[39].

To re-train Cellpose, we first created training datasets consisting of z-stacks of transmitted light images with the corresponding annotated images of uninfected erythrocytes and infected erythrocytes at the ring (10

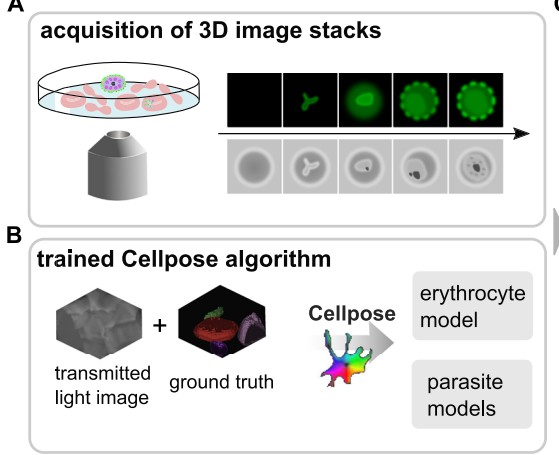
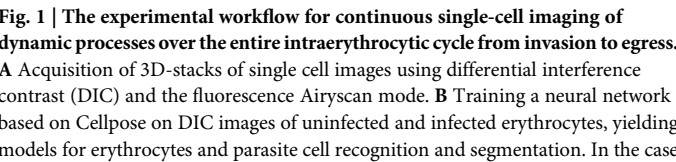
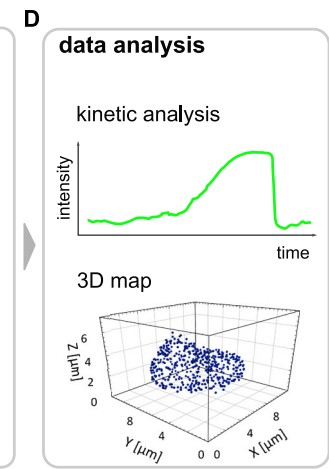

**Fig. 1 | The experimental workflow for continuous single-cell imaging of dynamic processes over the entire intraerythrocytic cycle from invasion to egress.** **A** Acquisition of 3D-stacks of single cell images using differential interference contrast (DIC) and the fluorescence Airyscan mode. **B** Training a neural network based on Cellpose on DIC images of uninfected and infected erythrocytes, yielding models for erythrocytes and parasite cell recognition and segmentation. In the case of infected erythrocyte, three parasite models were trained, on ring stages, trophozoites/schizonts and all stages. **C** Automatic cell segmentation via the trained models, delineating the erythrocyte plasma membrane, the erythrocyte cytosol and the parasite. **D** Analyzing the spatial and temporal dynamics of the process under investigation throughout the replicative cycle and 3D rendering of the captured images for visualization.

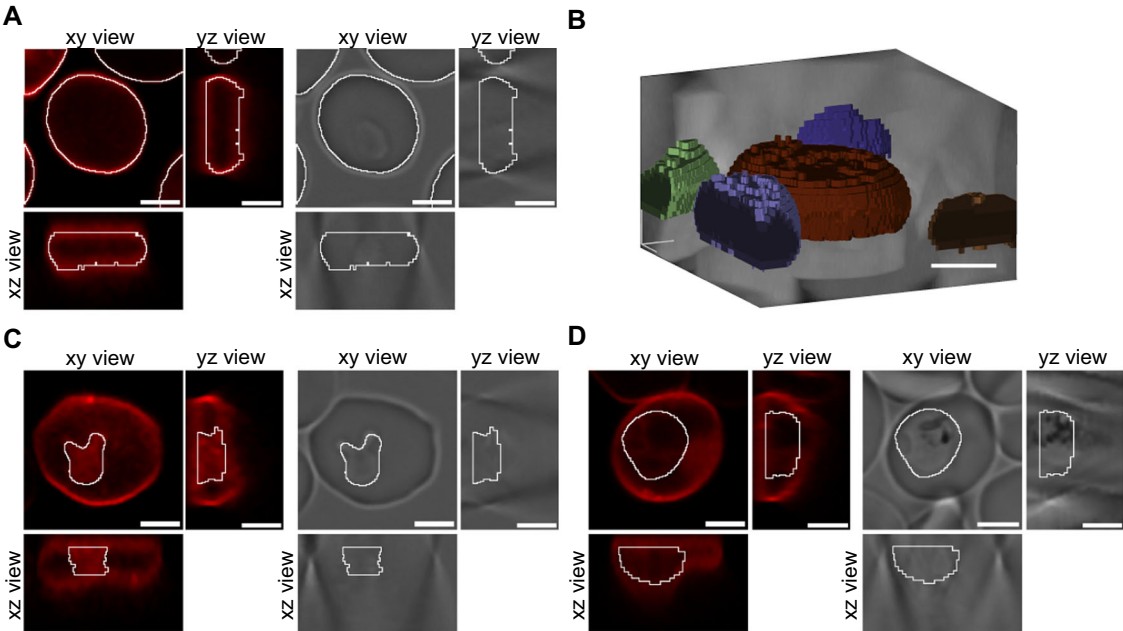

**Fig. 2 | Training dataset with ground truth annotation. A** Representative image showing erythrocytes stained with CellBrite to visualize the membrane. White lines outline the segmentation of the erythrocytes obtained by the ilastik carving workflow, which serves as ground truth. Dataset consists of n = 64 z-stacks with several cells. Top, confocal xy view, yz view. DIC xy view, yz view. Bottom, confocal xz view. DIC xz view. Scale bar, 3 µm. **B** Volume rendering of segmented erythrocytes. Scale bar, 3 µm Representative images showing infected erythrocytes stained with Cell-Brite to visualize the erythrocyte plasma membrane and the parasite inside the parasitophorous vacuole. White lines outline the segmentation of (**C**) a ring stage parasite and (**D**) a trophozoite, obtained by manual curation, which serves as ground truth. The dataset consists of n = 24 and n = 23 z-stacks of ring and trophozoite stage parasites, respectively. Scale bar, 3 µm.

to 18 h post invasion) and trophozoite/schizont stage (30 to 45 h post invasion). To facilitate the annotation process and discern better cell boundaries in 3D, we acquired confocal fluorescence images of cells stained with the membrane dye CellBrite Red. These confocal images were used for annotation purposes only and were not included in the training dataset. In a pre-processing step, the images were cropped to individual cells. For the erythrocyte training dataset, we imported the fluorescence image stacks into ilastik, an interactive machine learning-based tool for image analysis[40]. We employed the carving workflow in ilastik, which allows for volume segmentation based on boundary information in fluorescence or electron microscopy images. This segmentation approach yielded satisfactory results, considering the varying staining intensities of the erythrocytes (Fig. 2A and B). For the training datasets of infected erythrocytes, we imported the fluorescence image stacks into the Imaris software package. Each parasite was manually annotated using the surface rendering mode (Fig. 2C and D). Combined the training datasets comprised 111 3D stacks (64 uninfected erythrocytes, 23 ring stages, and 24 trophozoites/schizonts).

We next trained Cellpose in separate runs on the erythrocyte and the parasite datasets, the latter containing both rings and trophozoite/schizonts. We used the 3D extension of Cellpose, which operates similarly to the 2D version but incorporates xy, xz, and yz slices to construct a 3D gradient vector. All models were trained for 500 epochs and a 3.2-fold resolution increase in z-direction. As Cellpose exhibits lower performance for shapes with low convexity[39], which are particularly prevalent in the ring stages[41], we also trained Cellpose on rings only and on trophozoites/schizonts only. This yielded four training models: one for uninfected erythrocytes (erythrocyte model), one for ring stages (ring stage model), one for trophozoites/schizonts (late stage model), and a joint parasite model that incorporated both ring stages and trophozoites/schizonts.

### Evaluation of different Cellpose models

To evaluate the performance of each model we did 10-fold cross-validation, by splitting the annotated dataset into 10 equal-sized groups. Each model was then trained and evaluated 10 times, each time using a different group as the validation set and the remaining 9 groups as the training set. We evaluated the accuracy of each model by computing the average precision metric (AP) at different intersection-over-union (IoU) thresholds. The AP considers the number of true positives (TP), false positives (FP), and false negatives (FN) detections at a given IoU threshold and is calculated using the equation: $AP = TP/(TP + FP + FN)$. $AP_{0.5}$-values (AP for an IoU of 0.5) ranging from 0.54 to 0.95 were obtained for the different models, with the joint parasite model having the lowest value (Fig. 3).

We further evaluated the performance of the models by calculating the area under the curve (AUC) after averaging the results obtained from the 10 iterations. A high AUC value indicates a plausible model. For the erythrocyte model, this resulted in an AUC of 0.81 for the prediction of the entire red blood cell. For predicting the erythrocyte membrane only, we computed the difference between the segment dilated by one pixel and the segment eroded by one pixel, both in all directions. This process was applied to both the ground truth and the predictions, allowing us to compare the resulting shells at different accuracy thresholds. The same metric as before was employed to evaluate the performance, yielding an AUC of 0.63 for the shells. We observed marginal improvement in AUC for both cases (up to a maximum of 0.01) after implementing post-processing techniques, including filtering by size, merging of touching segments, and 2D smoothing (see materials and methods). The decrease in performance for the membrane segmentation can be attributed to the limited comparison of only the outer shell, which can exhibit greater variation in individual pixels while disregarding the matching volume entirely.

The three parasite models were evaluated separately on the ring stage dataset and the late stage dataset. The joint model achieved an AUC of 0.42 on the ring stage dataset and 0.61 on the late stage dataset. The model trained on the ring stage images alone performed poorly on the ring stage dataset (AUC = 0.33), while the specialized late stage model achieved an AUC of 0.70 on the late stage dataset. As a post-processing step, we merged all parasite detections within the relevant cell and smoothed the parasite volume by dilating by one pixel in all directions and eroding by one pixel in the x,y direction. This post-processing step improved the AUC for the joint model on the ring stage dataset from 0.42 to 0.49. However, it had a lesser impact on the late-stage dataset (AUC increased from 0.61 to 0.64). In the

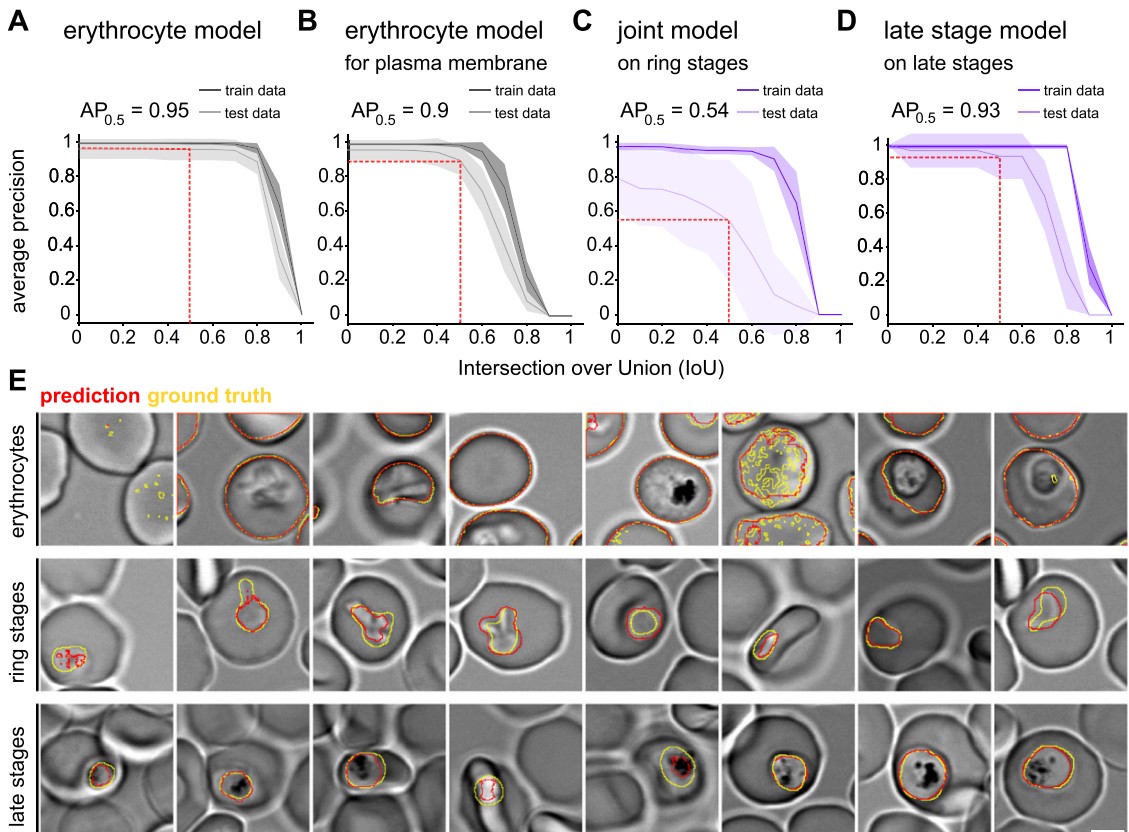

**Fig. 3 | Evaluation of different Cellpose models.** Shown are average precisions scores ± SD as a function of the intersection over union (IoU) threshold of the following models: (**A**) erythrocyte model; (**B**) erythrocyte model used to obtain the erythrocyte plasma membrane; (**C**) joint parasite model (trained on rings and trophozoites/schizonts) on ring stages; (**D**) late stage model (trained on trophozoites/schizonts) on late stages. (**A–D**) Dark color, evaluation on training dataset; light color, evaluation on test data. AUC, area under the curve. $AP_{0.5}$, average precision at 0.5 IoU, indicated with the dashed red line. **E** Representative xy slices depicting infected and uninfected erythrocytes at various depths within different z-stacks. DIC image with ground truth shown with yellow line and predicted masks shown with red line. Top row, erythrocyte model. Middle row, joint model on ring stages. Bottom row, late-stage model on late stages. Scale bar, 3 μm.

case of the single late-stage model, post-processing led to a decrease in AUC by 0.03. Despite the small difference in the performance of both models for the late stage dataset, we noticed in a qualitative analysis that the late stage model outperformed the joint parasite model. We, therefore, decided to use the joint model to predict the ring stage parasites and the late stage only model to predict the late-stage parasites.

**Proof of concept of segmentation strategy**

We next investigated the kinetics of KAHRP protein export as a proof of concept of our segmentation method. As mentioned above, KAHRP is a virulence factor that contributes to knob formation and, hence, to the disease-causing cytoadherence of infected red blood cells in the microvasculature[42,43]. KAHRP has to cross two membranes, the parasite plasma membrane and the parasitophorous vacuole membrane, on its way to the erythrocyte plasma membrane, where it binds to components of the membrane skeleton and to parasite factors, including the main adhesins, collectively termed PfEMP1[44,45]. Previous research on the export of KAHRP and the assembly of knobs has been limited to static snapshots, lacking information regarding the dynamics and kinetics of the underlying processes[35,46].

To study KAHRP dynamics during intraerythrocytic development, we generated a mutant parasite line expressing a C-terminal fusion protein of KAHRP with the photoactivatable fluorescent protein mEOS3.2 (Fig. 4A), using CRISPR/Cas9 genome editing technology[47]. Five clonal lines were obtained (Fig. 4B), and the integration event was confirmed by sequencing of the entire KAHRP locus. Western analysis using a KAHRP-specific antibody revealed a specific hybridization signal of ~100 kD in the parental FCR3 parasite line and of ~125 kD in the two mutant lines tested (Fig. 4C). Among the generated clones, we selected clone B4 for further investigation.

B4 produced knobs, as demonstrated by scanning electron microscopy (Fig. 4D). However, the knobs were larger and sparser than those generated by the parental parasite line (diameter: 116 ± 23 nm versus 61 ± 19 nm; 78 cells and 63 cells from $n = 4$ independent experiments, $p < 0.0001$, two-tailed t-test) (density: 3.2 ± 1.4 μm$^{-2}$ versus 14.35 ± 6.4 μm$^{-2}$; 78 cells and 63 cells from $n = 4$ independent experiments, $p < 0.0001$, two-tailed t-test) (Fig. 4E and F).

Previous studies have shown that knob morphology and density can affect cytoadhesion efficiency[48–50]. To test whether the knobs of the B4 line are functional, we performed a wash-out adhesion assay using chondroitin-4-sulfate (CSA) coated on petri dishes. CSA is an established receptor for cytoadhesion of infected erythrocytes in the intervillous space of the placenta during maternal malaria[51]. To this end, equal amounts of uninfected erythrocytes and purified erythrocytes infected with trophozoites of B4 and FCR3 were allowed to settle on CSA coated petri dishes for 30 min before a wall shear stress of ~0.1 Pa was applied for 5 min. The number of cells in contact with the surface were counted before and after the washout. We counted 2333 ± 517 mm$^{-2}$ ($n = 3$), 1832 ± 1072 mm$^{-2}$ ($n = 3$), and 1830 ± 1025 mm$^{-2}$ ($n = 3$) cells for uninfected erythrocytes and infected erythrocytes of the B4 line and FCR3, respectively, before the washout. After washout, 2.4 ± 2.1%, 54.5 ± 27.7%, and 58.4 ± 35.9% of the cell remained attached (Fig. 4G). There were no statistical differences in cytoadhesion efficiency between B4 and the parental line FCR3 ($p = 0.9882$; Tukey's multiple comparison). These data indicate that B4 forms functional knobs, despite altered knob morphology and density.

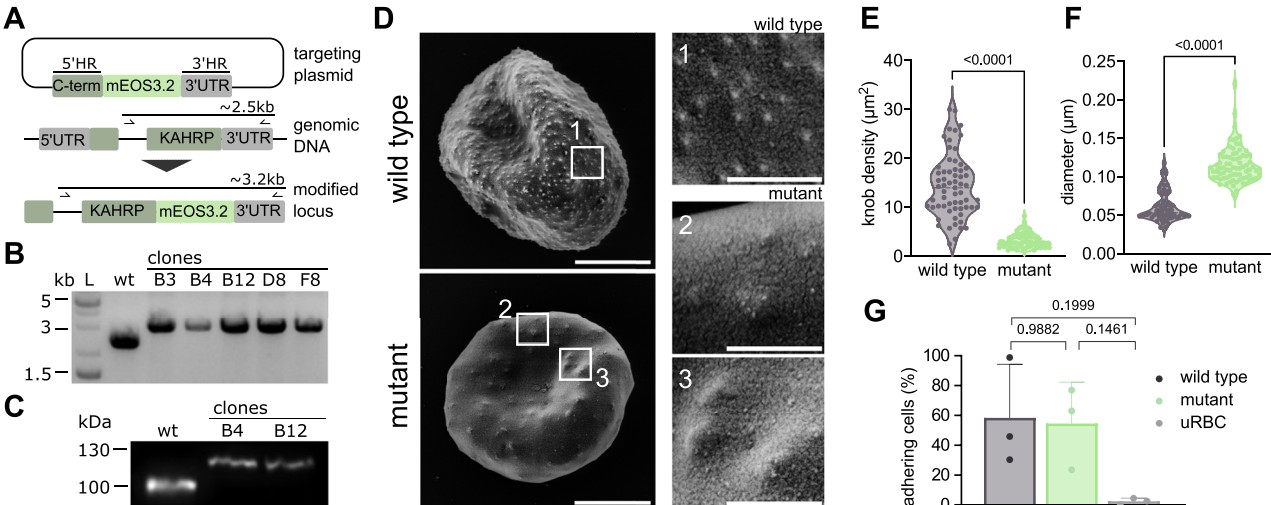

**Fig. 4 | Characterization of KAHRP::mEOS3.2 parasite line. A** Schematic illustration of CRISPR-based genome editing strategy to fuse the endogenous *P. falciparum kahrp* gene with the coding region of mEOS3.2. Arrows, primer binding sites. Not drawn to scale. **B** Pherogram showing products of diagnostic PCR using gDNA from the parental P. *falciparum* line FCR3 (WT) and five clonal mutants expressing a KAHRP::mEOS3.2 fusion protein. A DNA size marker is indicated. (for uncropped image, see Supplementary Fig. 1). **C** Western analysis confirming generation of KAHRP::mEOS3.2 in the two mutant clones investigated. Molecular masses in kilo Dalton (kDa). (for uncropped image, see Supplementary Fig. 2). **D** Scanning electron microscopy of intact erythrocytes infected with trophozoite stage parasites of FCR3 (top) and B4 (bottom). Scale bar, 2.5 μm. Panels on the right (1, FCR3; 2, 3, B4) show magnified views of the erythrocyte surface. Scale bar, 0.5 μm. **E** knob density and (**F**) knob diameter of FCR3 and B4. Data points represent individual infected erythrocytes (mean of 7 knob diameter measurements per erythrocyte). Dashed bold line, median; pointed lines, quartiles. Data from four independent biological experiments are shown, with *n* = 63 for FCR3 and *n* = 78 cells for B4. p < 0.0001, two-tailed t-test. **G** Cytoadhesion efficiency. Erythrocytes infected with FCR3 and B4 and uninfected erythrocytes as control were seeded on CSA-coated plastic dishes and subsequently washed. The number of adhering cells were normalized to the initial pre-wash number for each experiment. Each data point represents the average of five replicates, obtained by analyzing different sections of the CSA-coated plastic dish. The adhesion efficiency was comparable between FCR3 and B4. Error bars, SD; Tukey's multiple comparison.

We subsequently recorded the KAHRP fluorescence signal in single cells throughout the intraerythrocytic cycle, from invasion to egress, in intervals of 2 h under live cell imaging conditions, using a Zeiss Airyscan LSM900 microscope in super-resolution mode providing sub-diffraction resolution[52]. For each time point, a three-dimensional image was obtained.

We next applied our image analysis tools to the KAHRP dataset (Fig. 5A), however, we encountered new challenges not observed in the test dataset, as the test dataset only covered snapshots of different parasite stages. We found that cellular debris around the cell of interest as well as large hemozoin crystals and irregular patches in the cytosol posed difficulties for accurate detection and segmentation (Fig. 5B–H). Furthermore, we observed instances where the model failed to detect the bottom of the chamber when a large hemozoin crystal was close to it, resulting in protrusions in the z-direction. Consequently, we adapted the post-processing accordingly based on qualitative assessments of the predictions. The cell of interest was identified, and the associated segments were joined, based on size and increase in convexity, and stitched over time. Out of the 43 cells examined, five were excluded due to major detection errors at multiple time points. The remaining infected cells were analyzed for inaccuracies at each time point. In spite of post-processing, protrusion in the z-direction occurred in 83 out of a total of 1176 time points (Fig. 5B and G). Additionally, we identified larger missing fragments in 5 time points (Fig. 5B, E and F) and larger protrusions in 17 time points (Fig. 5B and D). Smaller missing parts were found in 81 time points, while smaller protrusions were found in 60 time points (Fig. 5B, E and C).

Overall, the erythrocyte model, along with the post-processing routine, exhibited satisfactory performance in predicting erythrocytes in the time-lapse dataset. From these predictions, we generated the membrane mask by eroding and dilating the predictions with a one-pixel margin. To identify and segment the parasite, we applied the parasite models to the time-lapse images. The ring stage was predicted using the joint model, while the late stage was predicted using the single late stage model. We successfully detected a parasite segment in 84.2% of all time points that showed an infected erythrocyte (Fig. 5I). However, the predicted parasite segment failed to accurately describe the parasite in cases of irregular light diffraction in the erythrocyte cytosol (Fig. 5H).

## Kinetics of KAHRP export

When analyzing the images, we noted that parasites originating from the same maternal schizont exhibited varying rates of development. Additionally, the cells displayed an extended average duration of the intraerythrocytic cycle, increasing from 46 ± 2 h under standard in vitro culture conditions to 62 ± 6 h under imaging conditions, despite being superfused with supplemented RPMI media at 37 °C. This variability in development rates among individual cells, combined with the overall extended intraerythrocytic cycle, introduced a new challenge in data analysis and interpretation. We addressed this limitation by normalizing the rate of development to that of an in vitro culture, using a method previously described by Grüring et al. [11] This method defined morphological criteria for each stage and each stage transition, such as the first visible hemozoin, hemozoin in a single food vacuole, the transition from dynamic parasite to static growth, a single large hemozoin spot, and movement of the food vacuole to a central position. Notably, imaging conditions affected all blood stages equally, as indicated by the comparison of each stage's contribution to the cycle duration (time-lapse: ring, 45.7%; trophozoite, 35.5%; schizont, 18.8%; cell culture: ring, 47.8%; trophozoite, 34.8%; schizont, 17.4%).

Using the normalized life cycle duration, KAHRP fluorescence appeared in the parasite ~8 ± 3 h post invasion, consistent with previous reports[44], and then increased in a sigmoidal fashion with time (Fig. 6A and B). Concurrently, the export to the erythrocyte cytosol began, reaching a steady state at around 30 h post invasion, when the parasite developed from the trophozoite to the schizont stage (Fig. 6B). Unlike previous reports, we did not detect a rim of fluorescence at the parasitophorous vacuolar membrane, a dotted pattern in the erythrocyte cytoplasm or an accumulation of KAHRP-associated fluorescence in the parasite's digestive vacuole[44]. KAHRP fluorescence accumulated at the erythrocyte membrane almost simultaneously with the cytosolic compartment and surpassed

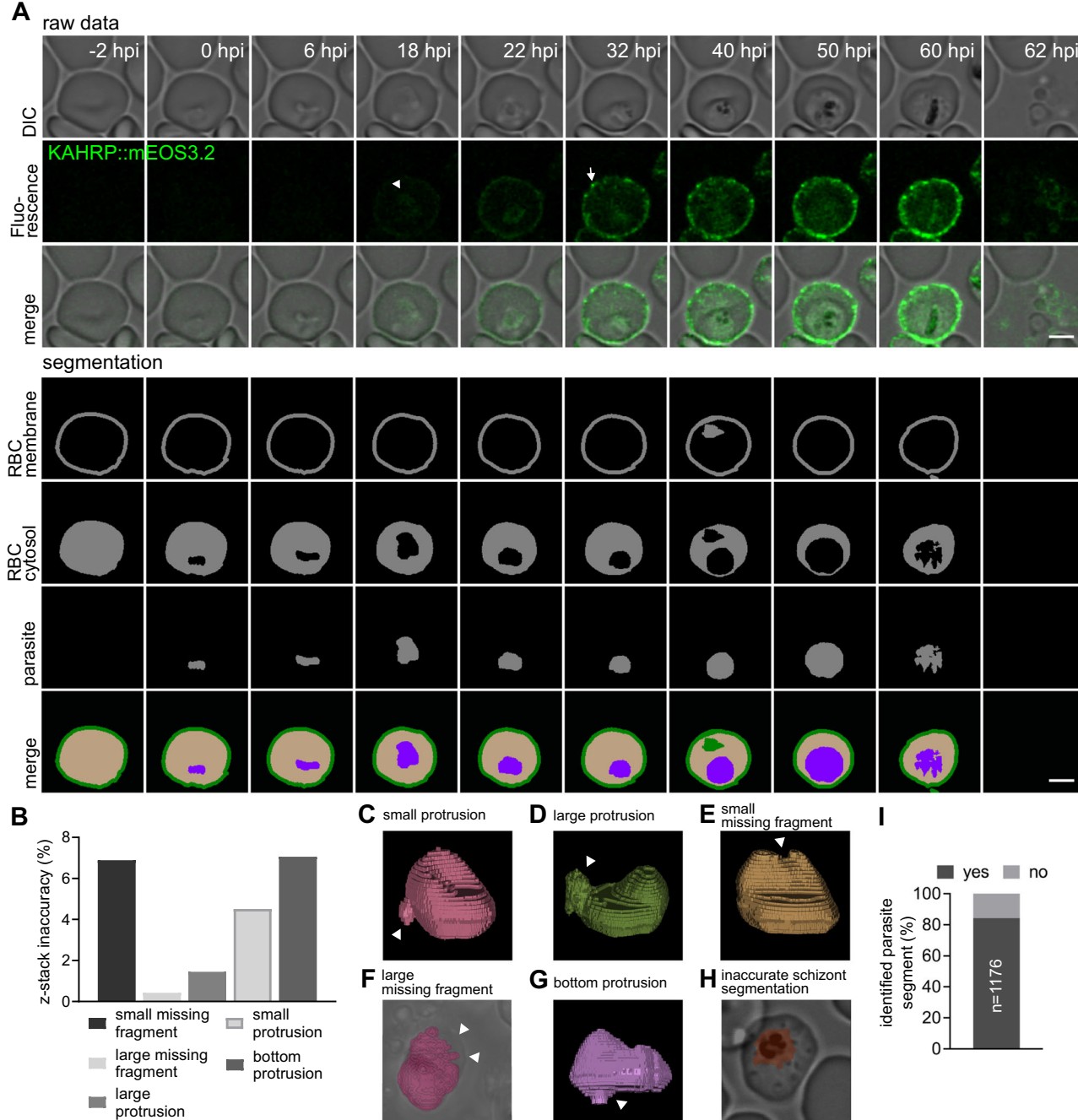

**Fig. 5 | Application of the segmentation strategy to continuous single-cell imaging of KAHRP export. A** Representative super-resolution time-lapse microscopic images of the B4 line expressing KAHRP::mEOS3.2 from invasion to egress. The time post invasion is indicated (hpi, hours post invasion). From top to bottom row: single z-slice DIC images; intensity projections of KAHRP::mEOS3.2 fluorescence images (generated using the Airyscan mode) summing several z-slices, with contrast being adjusted for better visualization. The white arrowhead indicates emerging KAHRP::mEOS3.2 accumulating in the parasite, The arrow indicates KAHRP fluorescence clusters at the erythrocyte membrane; DIC slice merged with KAHRP::mEOS3.2 fluorescence image (green); segmented red blood cell membrane; segmented red blood cell cytosol compartment; segmented parasite compartment; projection of segmented compartments, with red blood cell membrane (green), red blood cell cytosol (beige), and parasite (purple). Scale bar, 3 μm. **B** z-stack inaccuracies in erythrocyte segmentation after post-processing. A total of 1176 timepoints from 38 cells were investigated. **C–H** Various examples of inaccurate sample rendering (**I**) Percentage of parasite segments identified by the model of all time points showing a parasite. *n* = 1176 timepoints. Data from three independent experiments are shown.

cytosolic levels approximately 20 h post invasion. Membrane accumulation continued without saturation, ultimately accounting for ~53% of the total fluorescence intensity (Fig. 6A and B). At the erythrocyte membrane, KAHRP manifested as punctate structures that increased in intensity and quantity over time. Upon schizont rupture and merozoite egress, some KAHRP seemed to attach to the glass slide, while other molecules seemed to envelop the newly formed merozoites.

We next imported the 3D images of the segmented erythrocyte membrane into Imaris to identify associated KAHRP clusters. By adjusting the intensity threshold and cluster size, we optimized the detection to the smallest clusters, while minimizing false positives. This approach facilitated the visualization and quantification of membrane-proximate KAHRP clusters in 3D (Fig. 7A), which, in turn, allowed us to interrogate their dynamics and spatial organization. We found that the number of KAHRP

**Fig. 6 | Kinetics of KAHRP production and export.**
**A** KAHRP-associated fluorescence intensity ($F$ in arbitrary units) in the parasite, the erythrocyte cytosol compartment (RBC cytosol) and the erythrocyte membrane (RBC membrane) as a function of intraerythrocytic development. **B** KAHRP-associated fluorescence intensity ($F$ in arbitrary units) per $\mu m^2$ in the parasite, in the erythrocyte cytosol compartment (RBC cytosol) and the erythrocyte membrane (RBC membrane) as a function of intraerythrocytic development. **A, B** Note that the rate of development was normalized to that of an in vitro culture, using a method previously described by Grüring et al. (see Materials and Methods) [11]. The means ± SEM of n = 26 determinations are shown. A one-parameter sigmoidal function was fit to the data points (red lines).

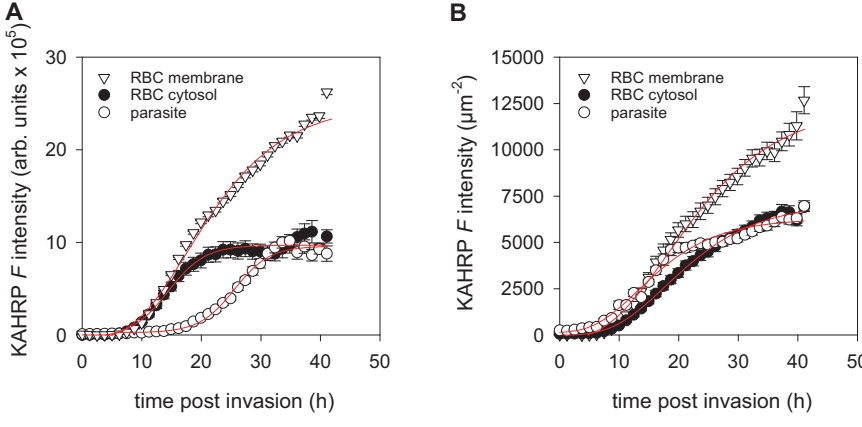

**Fig. 7 | Kinetics of KAHRP cluster formation at the erythrocyte plasma membrane.**
**A** Representative 3D projections showing formation of KAHRP clusters underneath the erythrocyte plasma membrane as a function of the adjusted time post invasion (hpi). The color code indicates fluorescence intensity. **B** Number of KAHRP clusters as a function of the normalized time post-invasion. Note that the rate of development was normalized to that of an in vitro culture, using a method previously described by Grüring et al. [11] The means ± SEM of 37 independent determinations (infected cells) are shown per time point. A one-parameter sigmoidal function was fit to the data points (red line). **C** Mean fluorescence intensity per KAHRP cluster as a function of the normalized time post-invasion (see above).

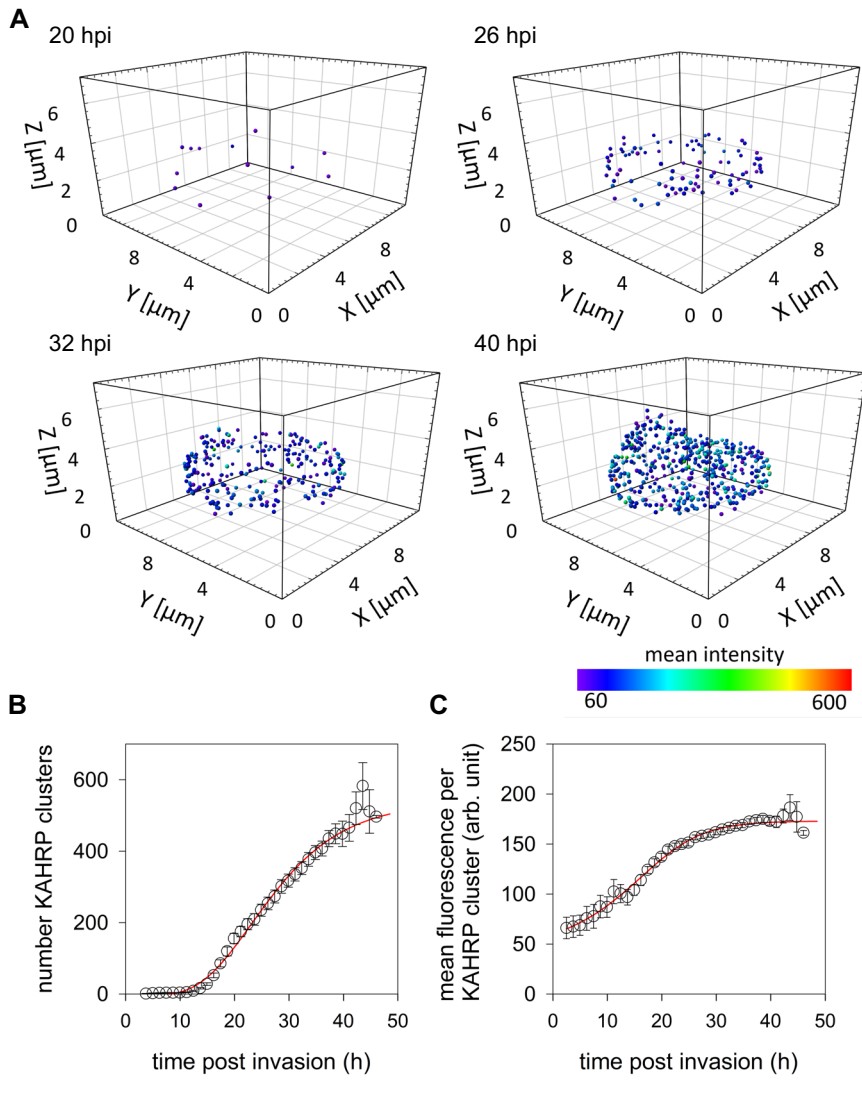

clusters increased in a sigmoidal fashion during intraerythrocytic development (Fig. 7B). Similarly, a sigmoidal relationship was observed between the mean fluorescence intensity per KAHRP cluster and the time post parasite invasion (Fig. 7C). These findings suggest that the parasite continuously forms new KAHRP clusters underneath the erythrocyte plasma membrane and that each cluster grows with time by recruiting new KAHRP molecules.

## Discussion

Here, we present a workflow designed for continuous recording and analysis of dynamic cellular processes at high resolution, specifically at the single-cell level in live *P. falciparum* blood stages. Our approach allows for the capture of the entire intraerythrocytic life cycle of *P. falciparum* in three dimensions and in a time-resolved manner, achieving high spatial and temporal resolution.

In configuring our system, we chose DIC images for their label-free nature, enabling imaging at low light intensity[53]. This approach preserves physiological conditions without requiring additional dyes or fluorescent markers. However, DIC images have limited contrast due to phase delay changes induced by the cells. In addition, DIC images lack quantitative intensity information[53], making image segmentation particularly challenging[20,54]. The segmentation task is further complicated by the anisotropic nature of DIC images and the large refractive index of erythrocytes.

To overcome these challenges, we employed Cellpose, a pre-trained convolutional neural network (CNNs) capable of learning valuable representations from raw input data through multiple layers of processing[20,55]. Cellpose, with its U-net architecture and 3D segmentation function, has demonstrated generalization capabilities for various tissue segmentation tasks[39]. Cellpose can be re-trained for specific tasks[39], and in our study, we leveraged this property to predict both the erythrocyte and the parasite, enabling the delineation of the erythrocyte plasma membrane, the erythrocyte cytosol, and the parasite compartment.

While the trained neural networks accurately segmented infected erythrocytes at the trophozoite and schizont stages, prediction accuracy was lower for ring stages, especially early ring-stage parasites. We attribute this to the lower convexity of ring stages[41] and challenges faced by the Cellpose segmentation method in recognizing them. Increased training data might enhance the prediction accuracy.

Despite the parasites remaining viable to develop and capable of re-infection throughout the observation period, the replicative cycle was extended. Achieving a normal 48-hour life cycle requires adjusting atmospheric tension, particularly $CO_2$ and $O_2$, to conditions resembling in vitro culture conditions, in addition to superfusion with 37 °C prewarmed supplemented RPMI 1640 media as used in this study.

It is important to note that our deep-learning algorithms were specifically trained on single-cell images recorded using an Airyscan microscope. The ability of our models to segment DIC images generated with another microscope has not yet been tested. Nevertheless, our approach highlights the utility of deep learning tools in solving challenging image analysis problems in *P. falciparum*. The implementation of deep learning image segmentation is particularly advantageous, facilitating high-throughput analysis of large amounts of image data.

As a proof of concept for our imaging and analysis workflow, we investigated trafficking of KAHRP into the host cell compartment. The *kahrp* gene is transcribed throughout the asexual blood stage development, with RNA levels peaking approximately 20 h post-invasion before subsequently decreasing to a minimum in late schizonts[56,57]. Initial KAHRP signals become discernible at the erythrocyte plasma membrane around 14 to 18 h post-invasion (time adjusted), consistent with previous reports[44]. Subsequently, the number of KAHRP clusters and their mean fluorescence intensity per cluster followed a sigmoidal increase. On the basis of these findings, we conclude that KAHRP clusters continuously form underneath the erythrocyte plasma membrane and that individual clusters grow by recruiting additional KAHRP molecules. These conclusions are consistent with previous atomic force microscopic studies demonstrating an age-related increase in knob density[36].

We acknowledge that these conclusions are based on a genetically engineered KAHRP::mEOS3.2 fusion protein. KAHRP acts as a modulator for the organization of membrane skeletal factors (e.g., spectrin, actin and ankyrin) and as a scaffold protein to assemble knobs, anchoring adhesins of the PfEMP1 family to the membrane skeleton[34,35]. Modifications of KAHRP through truncation, deletion, or insertion mutagenesis impact knob morphology and density[35,58]. In this investigation, parasites expressing the genomically encoded KAHRP::mEOS3.2 fusion protein exhibited larger and fewer knobs than the parental FCR3 parasite line. Evidently, fusing KAHRP at its C-terminus with mEOS3.2 altered the secondary and tertiary structure of the protein and, consequently, affected its function as a platform protein. However, we do not think that these limitations diminish the general conclusions drawn from the study.

Previous studies on trafficking of KAHRP observed accumulation of GFP-tagged KAHRP fusion proteins in the parasite ER, the digestive vacuole, and as a necklace-like pattern in the parasitophorous vacuolar lumen[44]. In contrast, we did not observe fluorescence clusters in compartments other than around the host cell plasma membrane. We attribute this discrepancy to differences in *kahrp* gene copy number and an overload of the protein export pathway in transiently transfected parasites with high numbers of the trans *kahrp* gene.

In conclusion, this paper highlights the importance and challenges of single-cell time-lapse imaging in malaria research and presents a novel approach based on deep learning image analysis and label-free images. By overcoming the technical challenges associated with imaging and analysis of large datasets, the small intracellular malaria parasite becomes more amenable to high-resolution 4D microscopy.

## Materials and Methods

### *P. falciparum* cell culture
The *P. falciparum* parasite cell lines (FCR3 and mutant) were cultured in $A^+$ erythrocytes in complete culture medium (RPMI 1640 supplemented with 2 mM L-Glutamine, 25 mM HEPES, 5% albumax, 5% $A^+$ human serum, 0.2 mM hypoxanthine and 20 µg ml$^{-1}$ gentamycin) under controlled atmospheric conditions of 3% $CO_2$, 5% $O_2$, 92% $N_2$ and 95% humidity at 37 °C as described[59]. The hematocrit and the parasitemia were maintained at 3.5% 1–6%, respectively. Parasites were selected for the knobby phenotype by gelatine floatation once or twice per week[60]. Cultures were synchronized to a window of 2 or 4 h using a combination of 0.5% sorbitol[61] and 100 µg ml$^{-1}$ heparin[62] or MACS purification[63].

### Sample preparation for live cell imaging
For live cell imaging, a synchronized parasite culture was washed twice with pre-warmed RPMI medium, and seeded on a Concanavalin A (1 mg ml$^{-1}$) coated (20 min at 37 °C) 35-mm ibidi chamber with glass bottom for 10 min at 37°C, as previously described[11,12]. The chamber was washed twice with imaging medium (RPMI 1640, w/o phenol-red (stable glutamine, 2 g l$^{-1}$ NaHCO$_3$), 25 mM HEPES, 0.2 mM hypoxanthine, 0.5% albumax, 12.5 µg ml$^{-1}$ gentamycin) until a faint red cell layer remained, and then filled with 5 ml of imaging medium. The chamber was incubated at 37 °C for 1.5–2 h for equilibration. Before imaging, the chamber was filled completely with imaging medium, and the lid was sealed with parafilm to prevent evaporation. For membrane staining, the parasite culture was incubated in imaging medium containing 2 µl ml$^{-1}$ CellbriteRed for 20 min at 37 °C in an orbital shaker, before seeding. For parasite staining, a schizont stage parasite culture was added to a culture of uninfected erythrocytes stained with CellbriteRed as described above. Subsequent invasion resulted in a stained parasitophorous vacuolar membrane surrounding the parasite.

### Super-resolution live cell imaging
Live cell imaging was conducted using point laser scanning confocal microscopy on a Zeiss LSM900 microscope equipped with an Airyscan detector and a Plan-Apochromat 63x/1.4 oil immersion objective. Imaging was carried out under temperature control at 37 °C. To ensure stability, the Definite Focus module was used for focus stabilization during image acquisition. Images were taken at multiple positions using an automated stage, and the intervals between captures were set at 2 h over a total period of 80 h. For multichannel imaging, we employed a sequential line scanning mode with 488 nm and 640 nm diode lasers, at 0.1% and 0.15% laser power. In stained samples, channels were not recorded sequentially but changed on each slice due to the rapid movement of the parasite. Brightfield images were obtained using a transmitted light PMT detector. GaAsP PMT and Airyscan detectors were used with gain adjustments ranging from 450 to 950 V. Images were acquired with a pixel size of 0.079 µm with Z-stack slices at 0.25 µm intervals, spanning a total range of 7 µm. The ZEN Blue 3.1 software was used for 3D Airyscan processing employing the automatically determined default Airyscan Filtering (AF) strength. The spatial resolution of the

Airyscan system is 160 nm in x and y, and 400 nm in z under the settings described herein[52].

## Training dataset

A training dataset was created, consisting of 64 z-stacks of erythrocytes and 47 z-stacks of parasites. Ground truth annotations were generated with the help of confocal fluorescent images. The stained erythrocytes were annotated using ilastik's carving pipeline resulting in three-dimensional binary masks representing the erythrocytes[40]. For the parasites within the erythrocytes, manual labeling, following the stained parasitophorous vacuolar membrane surrounding the parasite on the confocal fluorescence images, was performed using Bitplane's Imaris software. In instances where the parasite could not be clearly distinguished from the erythrocyte membrane in the upper z-planes, the labeling from the last visible z-plane was used to create a rounded structure in the subsequent two to three slices, matching the shape of the parasite. These labels from each plane were then merged to create three-dimensional binary masks representing the parasites as ground truths.

## Cellpose training and evaluation

The Cellpose cyto2 model was used for further specialization of the segmentation task[39]. For the erythrocyte segmentation task, a single model was trained to identify and segment erythrocytes. For the parasite segmentation task, because of the differences in shape between the ring and late-stage parasite, we trained three distinct models. One model was trained only on ring-stage parasites, the second model was trained only on late-stage parasites, and the third model was trained on all parasite images. Images underwent a 3.2-fold increase in resolution in z-direction by linear interpolation and the evaluation was performed using a 10-fold cross-validation to get performance estimates. We opted for cross-validation instead of a train/val/test split due to the deliberately small size of annotations used. This allowed us to use a larger part of the annotations for each training run (90%) and still get reliable test performance estimations by averaging the performance on the remaining 10% test data cross the 10 folds. Subsequently, we used all the data to train the final models. Each model was trained for 500 epochs. We used the standard training pipeline of Cellpose, which includes color normalization as well as random rotations and resizing as augmentations. To evaluate the model's performance, the average precision (AP) was computed for various Intersection over Union (IoU) thresholds, ranging from 0 to 1, for each test set. The AP values were then averaged across all the test sets and the area under the curve (AUC) was calculated. To obtain the AP scores, the number of true positive (TP), false positive (FP), and false negative (FN) detections at each IoU threshold were computed. The following post-processing routine was performed on erythrocytes: The cell of interest was determined as the largest segment within the 3D stack, possessing a size of at least 10,000 pixels after cropping the boundaries for 30 pixels in the x and y directions. The main segment was merged with neighboring segments unless there was a decrease in the volume-to-volume ratio of the main segment with a convex hull of more than 0.5% or if the main segment shared at least 450 pixels on its boundary with a smaller neighboring segment. This merging process was iterated until no further merging occurred. Each segment was then morphologically closed using a one-pixel radius ball in each xy-plane individually. Predictions were stitched together in the temporal direction when two segments in consecutive frames showed an IoU of at least 10%. The predictions were qualitatively assessed for the occurrence of missing fragments or protrusions at each timepoint using napari[64]. For post-processing of the parasite segment, all parasite detections within one cell were merged, and the parasite volume was smoothed by dilating by one pixel in all directions and eroding by one pixel in the x,y direction.

## Image analysis of time-lapse and KAHRP cluster

Time-lapse analysis: Qualitative assessment of time-lapse images was carried out using the ZEN Blue 3.1. software. Using the DIC images, cells were manually evaluated for parasite survival and health (red blood cells intact, regular progression of development stages). Further, the timepoint for the occurrence of parasite invasion, initial appearance of hemozoin, settlement of the previously dynamic parasites, movement of the hemozoin to a central position, and egress was manually determined. The following quantitative analysis was carried out using FIJI[65]. To analyze the accumulation of KAHRP::mEOS3.2 over time, the predictions obtained from cellpose were used to create binary segmentation masks. The membrane mask was created by eroding and dilating the erythrocyte prediction by 1 pixel each. The cytosolic mask was created by subtracting the membrane and parasite from the erythrocyte prediction. The background was determined for each cell by measuring the mean gray value of a region of interest near the infected erythrocyte and subtracted from the fluorescent signal in each plane of the 3D stack separately and at every time point. The raw integrated density of the fluorescent signal was determined at each time point in the area of the membrane, the cytosol, and the parasite. Data from individual cells were aligned by time of invasion.

KAHRP cluster analysis: 3D microscopic images were analyzed using the Imaris software (Bitplane) to identify and segment fluorescent spots representing the clusters of fluorescent KARHP signal. This analysis encompassed the complete temporal cycle of the data to assess the presence and characteristics of spots throughout the process. Spot detection was achieved using the "Spots" module within Imaris. To ensure accurate identification, parameters such as intensity thresholds and spot size were meticulously adjusted. The threshold was employed to distinguish true signals from background noise, set between 60 and 600 units in this study. Similarly, size filtering was applied to select objects with a diameter around 100 nanometers. This iterative process of parameter optimization was crucial for obtaining a reliable and representative population of spots for subsequent analysis, including downstream cluster analysis and interpretation of their spatial organization within the 3D data.

## Generation of KAHRP::mEOS3.2 parasite line

A fragment encoding mEOS3.2 was inserted in the frame after the KAHRP coding sequence, using genome editing transfected technology[47]. To this end, a fragment encoding the C-terminal KAHRP domain, the coding sequence of mEOS3.2, and 300 bp of the *kahrp* 3' untranslated region were cloned into the transfection vector pL6B[47], using a combination of In-Phusion cloning and classical ligase-mediated cloning of AflII and SacII restricted fragment. The guide RNA sequence was cloned into the pL6B vector at the BtgZI site using In-Phusion cloning. Transfections were performed by electroporation of 75 μg of each plasmid and the Cas9-expressing plasmid into a synchronized ring-stage culture of the *P. falciparum* line FCR3[47]. Transfected parasites were selected on 1.5 μM DSM1 (ENDO-THERM Life Sciences Molecules) and 5 nM of WR99210 (Sigma-Aldrich). Integration was confirmed by PCR of genomic DNA and sequencing analysis. Clones were obtained by limiting dilution and the integration was again confirmed by PCR and Western analysis, the latter using a custom-made rabbit peptide antibody against amino acids 288–302 of KAHRP (Eurogentec, dilution 1:2000) and a goat anti-rabbit secondary antibody (dilution 1:1000).

## Washout assay

Petri dishes were covered with 10 mg ml⁻¹ chondroitin-4-sulfate in PBS and incubated overnight at 4 °C. After washing, the dishes were treated with 1% BSA in PBS for one hour at 37 °C and then washed with imaging media. Parasites at the trophozoite stage (FCR3 wild type or mutant) were enriched using the MACS method[63], yielding >90% infected red blood cells. Enriched infected erythrocytes were returned to culture for 1 h. After removing the supernatant, cells were examined under a microscope and imaged as reference timepoint (five fields of view for each condition). Subsequently, the dishes were extensively washed with imaging media, and images were taken again (five fields of view for each condition). The images were analysed in a blinded manner using a randomization macro for counting red blood cells, and the results were normalized to the reference time point.

## Scanning electron microscopy

A trophozoite stage parasite culture was purified using the MACS method[63]. After fixation in 1% glutaraldehyde in 0.1 M cacodylate buffer for one hour or overnight at 4 °C, the cells were washed, and 20 µl of the resuspended cells were left on 0.01% Poly-L-Lysine incubated coverslips for 20 min at RT. Subsequently, the coverslips were stained with 1% osmium tetroxide for one hour at 4 °C and underwent a dehydration series with acetone (10 min each in 25%, 50%, 75%, 95%, 100% acetone at RT). The samples were dried using a critical point dryer and sputtered with 5 nm of palladium gold. Images were acquired using a Zeiss Leo 1530 scanning electron microscope at 2 kV and a working distance of $4 \pm 0.4$ nm using the SE2 detector. Knob density was determined by counting knob-like structures in four areas of 1 µm² per infected red blood cell, and the mean knob diameter was calculated from measurements of seven randomly chosen knobs. In total, 63 and 78 cells were analyzed for wild type and mutant, respectively.

## Statistics and reproducibility

Data were analyzed using Sigma Plot (v. 14.5, Systat) and GraphPad Prism 9. Statistical significance was determined using the two-tailed t-test or the analysis of variance test (ANOVA) followed by post hoc Tukey's test. $p$-values < 0.05 were considered significant. The number of independent biological replicates is indicated in the main test and/or the figure legends. Measurements obtained from an individual cell is considered an independent determination. For model validation, 1176 timepoints from 38 cells were investigated. If independent data points were averaged, then the mean ± the standard deviation (SD) or the standard error of the mean (SEM) is shown, as indicated in the text and/or figure legend.

## Reporting summary

Further information on research design is available in the Nature Portfolio Reporting Summary linked to this article.

## Data availability

All data supporting the findings of this study are available within the article and the Supplementary Data. Additionally, the images and segmentation underpinning this study are available at the zenodo public repository under https://doi.org/10.5281/zenodo.14281268[66].

## Code availability

The scripts including test examples are available at the githup public repository under https://github.com/sciai-lab/CP_RBC_Pfalciparum[67].

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

## Acknowledgements

We acknowledge support from the Infectious Diseases Imaging Platform (IDIP) at the Center for Integrative Infectious Disease Research, Medical Faculty, Heidelberg University. We thank A. Kernaja and M. Müller for technical assistance. This work was funded by the Deutsche Forschungsgemeinschaft (DFG, German Research Foundation) - project number 240245660-SFB 1129 (M.L., U.S.S, and F.A.H.), and the Ministry of Science, Research and Arts Baden-Württemberg, grant no: BW6-07 (M.L., D.G., P.W.).

## Author contributions

S.M.F., S.D., P.W., U.S.S., F.A.H., and M.L. designed the study. S.M.F., S.D., D.G., and C.P.S. performed experiments. S.M.F., S.D., D.G., C.P.S., P.W., U.S.S., F.H., and M.L. analyzed the results. S.M.F. and M.L. wrote the manuscript. All authors participated in the discussion and manuscript editing.

## Funding

## Competing interests

The authors declare no competing interests.
