## [Transparent Peer Review file · Communications Biology]

Deep learning image analysis for continuous single-cell imaging of dynamic processes in *Plasmodium falciparum*-infected erythrocytes

Corresponding Author: Professor Michael Lanzer

Version 0:

Reviewer comments:

Reviewer #1

(Remarks to the Author)

The performance of the parasite identification model is poor using Cellpose, and do not provide additional advantage.

Reviewer #2

(Remarks to the Author)

Brief summary of the manuscript

The authors describe a microscopy and image analysis workflow to study the *Plasmodium falciparum* blood stages using time-lapse and high-resolution imaging with low phototoxicity. Previous analysis approaches were limited to snapshots of multiple cells, missing dynamic cellular processes. The work presented shows how to effectively overcome the major analysis challenges with the small size and photosensitivity of the parasite as well as the difficulty to image the host erythrocytes.

The work described falls into three major parts:

- 1) Continuous, single-cell monitoring of live *P. falciparum* using three-dimensional differential interference contrast (DIC) and fluorescence microscopy with Airyscan.
- 2) Automated cell segmentation through pre-trained deep-learning algorithms
- 3) 3D rendering for visualization and time-resolved analyses

The authors provide a proof of concept analysis to validate the workflow.

Overall impression of the work

The authors very well introduce the analysis challenges and how the workflow addresses them. Overall their method is well motivated and introduced, outlining an important research problem the analysis method supports. Most importantly there is in-depth validation of the core aspect of the workflow around the 3D image segmentation with CellPose. I particularly enjoyed how the authors clearly explained the issues they encountered and how they addressed them. Overall the limitations of the workflow are discussed. However, a detailed introduction and discussion in contrast to other established analysis approaches is lacking. There is a proof of concept analysis. Although the analysis presentation with the figures should be improved throughout the study. My major concern is mostly with the detailed method description as well as the accessibility of the workflow and analysis. Here particularly the image analysis description is insufficient, especially since no examples and scripts are provided in the supplements and/or a public repository.

Specific comments, with recommendations for addressing each comment

I walk through the individual sections in the results and methods with summaries of my review followed by my recommendations.

A neuronal network for segmentation of *P. falciparum* infected erythrocytes

Looks to be a good workflow setup using state of the art methods to practically and pragmatically address a very interesting analysis problem. Figure 1 can be improved in clarity.

Minor: Figure 1 can be easily improved by labeling the different workflow parts with (A,B,C etc). Proper description of the Figure legends and references to the workflow parts in the text.

Evaluation of different Cellpose models

Model validated with 10-fold cross validation. No comparison with other methods. But quantitative validation seems sufficient to assess the validity of the overall segmentation approach. I like that different examples are shown contrasting with the ground truth. Showing transparently good segmentation but also bad ones. Well done! The information is very difficult to read in Figure 3. Needs to improve.

Major: Figure 3 – the labels – prediction vs ground truth are hard to read. At the first pass I did not see them. Also only at the second pass I recognized that there are white outlines in the image. This needs to be improved. Please use different colors for the outlines such that the important information becomes clearly visible.

Proof of concept of segmentation strategy

Introduction explains problem well: KAHRP protein export dynamics during intraerythrocytic development. The authors went to great length to make a good validation of the underlying biological system they use for their proof of concept. Graphs for analysis are clear and the flow of the text is good. The discussion of previous approaches is a bit sparse.

Really like how the authors are transparent with the new problems they encountered during the application on their data and how they address them! Very good work. Figure 5 is a bit confusing to me and I have a hard time completely understanding the message at the first read.

Minor: Figure 4D is confusingly labeled and its not immediately clear what is wild type, mutant and where the insets are coming from. Can be easily improved by using clear figure labels.

Minor: Figure 5A could be easily improved by labeling the rows in the figure directly.

Kinetics of KAHRP export

Comprehensive analysis that is well presented.

Discussion

Limitation of analysis for transfer to other microscopy setups are discussed. Great that the authors address this point. I am not happy that all the Data is only available upon request. This is a methods paper so in order for this work to have impact in the field, the method should be somewhat accessible, ideally via a public repository. I don't expect a full fledged plugin or that all ground truth is available. So it would be great to have a very basic workflow available in a public repository (e.g. Zenodo):

- Some example or validation data
- Analysis scripts and results to allow to walk through analysis

Of course this now raises concerns if the overall methods description is good enough for reproduction. Thus I will continue my review deeper in the methods.

Major: Make part of the proof of concept accessible in a repository. With a limited amount of example inputs, analysis scripts and results.

Methods

Image analysis and statistics

The description of the image analysis methods is insufficient. In the introduction section the authors mention they use 3 different software platforms for image analysis (Zen Blue, Fiji and napari) however it is unclear how they were used. The description of the analysis concerning the accumulation of KAHRP::mEOS2.2 over time is a bit better. But here they mention only the software Imaris. Not the previously introduced software platforms. There are only statements concerning the manual analysis of parasite properties without actual information on how this was achieved. The level of detail provided insufficient.

Major: Outline image analysis methods clearly, with step by step descriptions of when which software has been used with what function. More details need to be provided particularly when it comes to the manual evaluations that are mentioned. Manual analysis is still using visual cues and criteria that should be described transparently. Alternatively provide example data.

Summary of requested revisions:

Major: Figure 3 – the labels – prediction vs ground truth are hard to read. At the first pass I did not see them. Also only at the second pass I recognized that there are white outlines in the image. This needs to be improved. Please use different colors for the outlines such that the important information becomes clearly visible.

Major: Make part of the proof of concept accessible in a repository. With a limited amount of example inputs, analysis scripts and results.

Major: Outline image analysis methods clearly, with step by step descriptions of when which software has been used with what function. More details need to be provided particularly when it comes to the manual evaluations that are mentioned. Manual analysis is still using visual cues and criteria that should be described transparently. Alternatively provide example data.

Minor: Figure 1 can be easily improved by labeling the different workflow parts with (A,B,C etc). Proper description of the Figure legends and references to the workflow parts in the text.

Minor: Figure 4D is confusingly labeled and its not immediately clear what is wild type, mutant and where the insets are coming from. Can be easily improved by using clear figure labels.

Minor: Figure 5A could be easily improved by labeling the rows in the figure directly.

Version 1:

Reviewer comments:

Reviewer #1

(Remarks to the Author)

The authors have tried to improve the manuscript by addressing many of the concerns in the paper, especially by refining the figures and uploading the experimental workflow in a public repository such as GitHub. They have only selected one major concern from my comments regarding the poor performance of the developed method in identifying ring-stage parasite and have completely ignored my other concerns. They have however addressed some minor concerns that I raised in the first round of revision.

The authors have failed to cite relevant references especially in relation with the application of machine learning and artificial intelligence in the identification of parasites including their different stages during infection which is a central part of this study. They must cite similar studies, especially the most recent ones including the papers from the group of Prof. Janos Torok (see comments from round 1) and compare this study with the findings from previous studies. This would be required to understand whether there is a significant advance in technology or not.

They have not addressed my concerns regarding their image acquisition settings, and Zeiss Airy Scan 3D acquisition parameter Z-step size of 0.25 nm (Is this even possible?). They have not mentioned anything regarding the resolution of the system although they mention about high spatial and temporal resolution. Spatial resolution can be improved with Airy Scan by about two-fold, but I am not sure whether this is the case here. Do they have a comparison somewhere as a supplementary figure to prove this point?

Further, temporal resolution is a term that has been used in the introduction and abstract, but acquisition of images was done every two hours – a time interval sufficient for several biological events to complete. This is not a major problem in conventional fluorescence microscopes, including confocal or widefield microscopy. They also need to cite the paper where lattice light sheet and structure illumination microscopy was used to study plasmodium invasion of erythrocytes (Geoghegan, Niall D., et al. "4D analysis of malaria parasite invasion offers insights into erythrocyte membrane remodeling and parasitophorous vacuole formation." *Nature communications* 12.1 (2021)

1) In Figure 4G, with the number of independent experiments being only 3, it is not advisable to carry out statistical tests. It should be at least $n=5$ (Curtis, Michael J., et al. "Experimental design and analysis and their reporting II: Updated and simplified guidance for authors and peer reviewers." *British journal of pharmacology* 175.7 (2018). Although, it can be included for the purpose of proving the point in an experimental observation.

2) In Figure 7, mean number of KAHRP clusters seems to be increasing and mean fluorescence of each KAHRP cluster is also increasing. Did you check the KAHRP cluster size? Explain why this was not included. This can be easily obtained using Imaris. They should also compare this feature with the previous observation where they have seen necklace bead-like patterns for KAHRP. The spatial distribution features of KAHRP would be a useful addition in the paper.

Hence, my other concerns still stand. They need to look at the comments from the first round and address them accordingly. I am not copy pasting it here.

The only novelty in the paper lies in the fact that it uses 3D-DIC images at every time point for segmentation and identification of parasites, however, whether this technique is superior to other methods reported previously is still a point of contention.

Reviewer #2

(Remarks to the Author)

I have reviewed the revised paper. The authors fully addressed my concerns.

Version 2:

Reviewer comments:

Reviewer #1

(Remarks to the Author)

I am quite happy with the new changes in the manuscript. The novelty lies in the label-free detection of various stages of malarial parasites using DIC images. However, caution should be exercised while comparing their paper with others, particularly with the one by Geoghegan, Niall D., et al. (Nature communications, 2021) which uses lattice light sheet, a technique that uses low dose of light. Hence, emphasis should be on the label-free DIC technique rather than on the phototoxicity of the light in the introduction and discussion. This is important because this paper has recorded DIC and fluorescence images simultaneously, for example while studying KAHRP-mEOS dynamics. This contradicts the persistent claims of negligible phototoxicity with this approach.

Also, the rebuttal has not addressed the necklace-like pattern seen for KAHRP in previous studies. It is advisable to address this in the discussion.

Furthermore, the rebuttal states the XY resolution as 160 nm in comment 2, then goes on to state it as 140 nm in comment 5. Nonetheless, the generalisation remains valid, especially in their novel use of the DIC-based label-free approach in identifying parasites.

Responses to reviewers' comments on manuscript COMMSBIO-24-3814

We thank the reviewers for their encouraging comments and helpful suggestions.

Reviewer #1: The performance of the parasite identification model is poor using Cellpose, and do not provide additional advantage.

We appreciate the reviewer's feedback regarding the model's performance and agree that the segmentation models described in this study do not yet achieve absolute predictive accuracy. However, we respectfully disagree with the characterization of their performance as "poor." As demonstrated in Figure 3 and detailed in the manuscript, the model for detecting and segmenting trophozoites and schizonts performs with high accuracy, achieving an average precision metric (AP_{50}) of 0.93. The model for identifying and segmenting ring stages, while not as strong, still achieves a reasonable precision level, with an AP_{50} of 0.54. We achieved this performance with extremely limited training set sizes of only 24 segmented ring stages and 23 segmented trophozoites and schizonts. Segmenting the ring stages is challenging because of their varied shapes. We acknowledge there is room for improvement, particularly for the ring stage model, and have addressed this in the manuscript (see Discussion section on page 13). However, for many applications, including those presented in this study, an AP_{50} of 54% for ring stages and 93% for trophozoites and schizonts is both practical and highly beneficial for the efficient analysis of large datasets. To strengthen further our argument, we have revised the former Supplementary Figure 1 and moved it to the main manuscript, new Figure 6, which now shows the time course of KAHRP fluorescence accumulation in different subcellular compartments (parasite, erythrocyte cytosol and erythrocyte plasma membrane) throughout the 48 hour replicative cycle of the parasite.

Reviewer #2: Brief summary of the manuscript

The authors describe a microscopy and image analysis workflow to study the *Plasmodium falciparum* blood stages using time-lapse and high-resolution imaging with low phototoxicity. Previous analysis approaches were limited to snapshots of multiple cells, missing dynamic cellular processes. The work presented shows how to effectively overcome the major analysis challenges with the small size and photosensitivity of the parasite as well as the difficulty to image the host erythrocytes.

The work described falls into three major parts:

- 1) Continuous, single-cell monitoring of live *P. falciparum* using three-dimensional differential interference contrast (DIC) and fluorescence microscopy with Airyscan.
- 2) Automated cell segmentation through pre-trained deep-learning algorithms
- 3) 3D rendering for visualization and time-resolved analyses

The authors provide a proof of concept analysis to validate the workflow.

Overall impression of the work

The authors very well introduce the analysis challenges and how the workflow addresses them. Overall their method is well motivated and introduced, outlining an important research problem the analysis method supports. Most importantly there is in-depth validation of the core aspect of

the workflow around the 3D image segmentation with CellPose. I particularly enjoyed how the authors clearly explained the issues they encountered and how they addressed them. Overall the limitations of the workflow are discussed. However, a detailed introduction and discussion in contrast to other established analysis approaches is lacking. There is a proof of concept analysis. Although the analysis presentation with the figures should be improved throughout the study. My major concern is mostly with the detailed method description as well as the accessibility of the workflow and analysis. Here particularly the image analysis description is insufficient, especially since no examples and scripts are provided in the supplements and/or a public repository.

We thank the reviewer for the positive and encouraging feedback. We appreciate the recognition of the challenges and motivations behind our workflow, as well as the core validation around 3D image segmentation with CellPose.

We understand and acknowledge the reviewer's concerns regarding the level of detail in the method description and apologize for the oversight in not depositing the scripts in a public repository. To address this concern, we have now uploaded all relevant scripts, including test examples, to GitHub (https://github.com/sciai-lab/CP_RBC_Pfalciparum), making them fully accessible for further examination and replication. Additionally, we have uploaded all images and segmentations underpinning this study to Zenodo (<https://doi.org/10.5281/zenodo.14281268>) and have attached a source data file to the manuscript compiling the quantitative data supporting the graphs.

We have further enhanced the quality and clarity of the figures throughout the manuscript. We have also expanded the introduction to provide a more comprehensive comparison with established analysis approaches, particularly in the context of image segmentation for malaria research and related fields.

Specific comments, with recommendations for addressing each comment

I walk through the individual sections in the results and methods with summaries of my review followed by my recommendations.

A neuronal network for segmentation of *P. falciparum* infected erythrocytes

Looks to be a good workflow setup using state of the art methods to practically and pragmatically address a very interesting analysis problem. Figure 1 can be improved in clarity. Minor: Figure 1 can be easily improved by labeling the different workflow parts with (A,B,C etc). Proper description of the Figure legends and references to the workflow parts in the text.

We thank the reviewer for their constructive feedback. In response, we have improved Figure 1 by labeling the different parts of the workflow (A, B, C, etc.) as suggested. We have clarified the figure legend to enhance understanding and included explicit references to each part of the workflow within the main text. Additionally, we have increased the resolution of the images to improve overall clarity.

Evaluation of different Cellpose models

Model validated with 10-fold cross validation. No comparison with other methods. But quantitative validation seems sufficient to assess the validity of the overall segmentation approach. I like that different examples are shown contrasting with the ground truth. Showing transparently good segmentation but also bad ones. Well done! The information is very difficult to read in Figure 3. Needs to improve.

Major: Figure 3 – the labels – prediction vs ground truth are hard to read. At the first pass I did not see them. Also only at the second pass I recognized that there are white outlines in the image. This needs to be improved. Please use different colors for the outlines such that the important information becomes clearly visible.

We thank the reviewer for this valuable feedback. To improve the clarity of Figure 3, we have added descriptive labels directly within subfigures 3A to 3D. In subfigure 3E, we have adjusted the colors of the outlines to yellow for the ground truth segmentation and red for the predictions, providing a clearer contrast against the grayscale images and enhancing the visibility of key information.

Proof of concept of segmentation strategy

Introduction explains problem well: KAHRP protein export dynamics during intraerythrocytic development. The authors went to great length to make a good validation of the underlying biological system they use for their proof of concept. Graphs for analysis are clear and the flow of the text is good. The discussion of previous approaches is a bit sparse. Really like how the authors are transparent with the new problems they encountered during the application on their data and how they address them! Very good work. Figure 5 is a bit confusing to me and I have a hard time completely understanding the message at the first read. Minor: Figure 4D is confusingly labeled and its not immediately clear what is wild type, mutant and where the insets are coming from. Can be easily improved by using clear figure labels. Minor: Figure 5A could be easily improved by labeling the rows in the figure directly.

We appreciate the reviewer's suggestions to improve the clarity of Figures 4 and 5. In Figure 4D, we have rearranged the layout and highlighted the area from which the insets were derived with a white box in the SEM overview, making it clearer where the wild-type and mutant sections are located. For Figure 5A, we have added row labels directly to the figure as recommended. Additionally, we have included descriptive text for Figures 5C-H and clarified missing details in the figure legend. We hope these enhancements address the reviewer's concerns and improve the figures' clarity.

Kinetics of KAHRP export

Comprehensive analysis that is well presented.

Discussion

Limitation of analysis for transfer to other microscopy setups are discussed. Great that the authors address this point. I am not happy that all the Data is only available upon request. This is a methods paper so in order for this work to have impact in the field, the method should be somewhat accessible, ideally via a public repository. I don't expect a full fledged plugin or that all ground truth is available. So it would be great to have a very basic workflow available in a public repository (e.g. Zenodo):

- Some example or validation data
- Analysis scripts and results to allow to walk through analysis.

Of course this now raises concerns if the overall methods description is good enough for reproduction. Thus I will continue my review deeper in the methods. Major: Make part of the proof of concept accessible in a repository. With a limited amount of example inputs, analysis scripts and results.

We apologize for the oversight in not making the scripts and test examples available in a public repository. We have now addressed this by providing a repository on GitHub, which includes example inputs, analysis scripts, and results for the CellPose fine-tuning and segmentation proof-

of-concept. The repository can be accessed using the following link: https://github.com/sciai-lab/CP_RBC_Pfalciparum. Thank you for this suggestion, which we believe will enhance the accessibility and reproducibility of our work.

Methods

Image analysis and statistics

The description of the image analysis methods is insufficient. In the introduction section the authors mention they use 3 different software platforms for image analysis (Zen Blue, Fiji and napari) however it is unclear how they were used. The description of the analysis concerning the accumulation of KAHRP::mEOS2.2 over time is a bit better. But here they mention only the software Imaris. Not the previously introduced software platforms. There are only statements concerning the manual analysis of parasite properties without actual information on how this was achieved. The level of detail provided is insufficient. Major: Outline image analysis methods clearly, with step by step descriptions of when which software has been used with what function. More details need to be provided particularly when it comes to the manual evaluations that are mentioned. Manual analysis is still using visual cues and criteria that should be described transparently. Alternatively provide example data.

We apologize for the lack of clarity and appreciate the reviewer's emphasis on transparency and reproducibility. To address these concerns, we have restructured the "Materials and Methods" section on image analysis into several distinct subsections, detailing each step and specifying the software and functions used. The updated section now includes separate descriptions for each stage: training dataset preparation, CellPose analysis, time-lapse image analysis, and KAHRP cluster analysis.

Each software tool and its specific role in the workflow is now clearly outlined in the relevant subsection. For example, the training dataset was created using Ilastik and Imaris, while CellPose segmentations were qualitatively evaluated with Napari. We have added further details on the visual cues used for the manual evaluations, which were conducted in ZEN Blue for qualitative assessments and in Fiji for quantitative analysis. KAHRP cluster analysis was conducted using Imaris.

Thank you for highlighting this need, and we trust these updates will improve the transparency and reproducibility of our methods.

Summary of requested revisions:

Major: Figure 3 – the labels – prediction vs ground truth are hard to read. At the first pass I did not see them. Also only at the second pass I recognized that there are white outlines in the image. This needs to be improved. Please use different colors for the outlines such that the important information becomes clearly visible.

Done as suggested, please see above.

Major: Make part of the proof of concept accessible in a repository. With a limited amount of example inputs, analysis scripts and results.

The scripts and test examples are now available at Github under the following link: https://github.com/sciai-lab/CP_RBC_Pfalciparum

The images and segmentations underpinning this study are available at Zenodo under the following link: <https://doi.org/10.5281/zenodo.14281268>

Additionally a source data file is included, in which the quantitative data underpinning graphs are compiled. By making all data and scripts publicly available, we aim to uphold the highest standards of data transparency and reproducibility.

Major: Outline image analysis methods clearly, with step by step descriptions of when which software has been used with what function. More details need to be provided particularly when it comes to the manual evaluations that are mentioned. Manual analysis is still using visual cues and criteria that should be described transparently. Alternatively provide example data.

We have improved the materials and methods section and now clearly explain at which step in the workflow a certain software tool was used (see also above).

Minor: Figure 1 can be easily improved by labeling the different workflow parts with (A,B,C etc). Proper description of the Figure legends and references to the workflow parts in the text.

Done as suggested by the reviewer (see above)

Minor: Figure 4D is confusingly labeled and its not immediately clear what is wild type, mutant and where the insets are coming from. Can be easily improved by using clear figure labels.

Done, as suggested by the reviewer (see above).

Minor: Figure 5A could be easily improved by labeling the rows in the figure directly.

Done, as suggested by the reviewer (see above).

Responses to reviewers' comments on manuscript COMMSBIO-24-3814A

Reviewer #2 (Remarks to the Author):

I have reviewed the revised paper. The authors fully addressed my concerns.

We are very grateful to the reviewer for accepting the manuscript in its present form.

Reviewer #1 (Remarks to the Author):

The authors have tried to improve the manuscript by addressing many of the concerns in the paper, especially by refining the figures and uploading the experimental workflow in a public repository such as GitHub. They have only selected one major concern from my comments regarding the poor performance of the developed method in identifying ring-stage parasite and have completely ignored my other concerns. They have however addressed some minor concerns that I raised in the first round of revision.

We are surprised by the reviewer's comments and their allegation that we largely ignored their suggestions and advise. We would like to respectfully clarify that we made a concerted effort to address all recommendations from the initial review process in an open and transparent manner. To ensure clarity and thoroughness, we included the reviewer's comments verbatim in our response letter and provided detailed replies to each point. However, it appears that only one comment from this reviewer was forwarded to us via the editorial office and the online submission portal. This single comment pertained to the performance of the ring stage model. Our responses to this concern, along with the changes made to the manuscript during the first revision cycle, seemed to have satisfied this reviewer.

Now the reviewer refers to several additional comments that were apparently intended for the first review cycle but never communicated to us. It seems that there was a breakdown in communication at some point, a regrettable mishap for which, however, we cannot be held responsible. In a communication with the Editor on this matter we were informed that "concerns that have not been sufficiently raised in the initial revision cycle, from the Editorial point of view, cannot be considered in later revision cycles".

Nonetheless, we feel it is important to address these newly raised concerns, as ignoring them would be against the spirit of constructive scientific dialogue. Therefore, we have decided to incorporate responses to these additional comments in a revised version of the manuscript.

Comment 1: The authors have failed to cite relevant references especially in relation with the application of machine learning and artificial intelligence in the identification of parasites including their different stages during infection which is a central part of this study. They must cite similar studies, especially the most recent ones including the papers from the group of Prof. Janos Torok (see comments from round 1) and compare this study with the findings from previous studies. This would be required to understand whether there is a significant advance in technology or not.

We apologize for the oversight, and now cite the work by the groups of Prof. Torok and Prof. Rogers in the introduction:

"In the context of *P. falciparum*, Preißinger et al. have recently demonstrated a neural network capable of identifying individual erythrocytes in multi-cellular two-dimensional images, distinguishing between infected and uninfected red blood cells, and classifying parasite stages into rings, trophozoites and schizonts³². In another study, Geoghegan et al. employed lattice-

sheet microscopy to investigate parasitophorous vacuolar formation (a compartment separating the parasite from the erythrocyte cytoplasm), capturing this process in a space and time-resolved manner during invasion ¹³." (end of page 4)

While the cited work introduces valuable tools for the identification of infected erythrocytes, classification of parasite stages, and 4D parasite imaging, it does not address the critical problem of segmenting images of infected erythrocytes captured under life-cell conditions, which maintain cell viability by limiting cell damage caused by light induced phototoxicity. This is the gap our study aims to fill. In our research, we use label-free DIC imaging, which allows imaging at low light intensity, thereby minimizing phototoxicity. Furthermore, we present a neural network-based tool specifically designed for the automated segmentation of DIC images of the infected erythrocyte. This tool overcomes inherent challenges associated with analyzing DIC images, such as limited contrast and the absence of quantitative intensity information. As a result, it enables the precise delineation of the parasite compartment and the erythrocytes compartment, including the erythrocyte plasma membrane, cytosol, and the boundaries of the parasite (see manuscript for details).

Comment 2: They have not addressed my concerns regarding their image acquisition settings, and Zeiss Airy Scan 3D acquisition parameter Z-step size of 0.25 nm (Is this even possible?). They have not mentioned anything regarding the resolution of the system although they mention about high spatial and temporal resolution. Spatial resolution can be improved with Airy Scan by about two-fold, but I am not sure whether this is the case here. Do they have a comparison somewhere as a supplementary figure to prove this point?

We apologize for the typographic error. The correct value is 0.25 μm (Material and Methods section, page 18, first paragraph).

Additionally, we now provide information regarding the spatial resolution of our microscopic set-up.

"The spatial resolution of the Airyscan system is 160 nm in x and y, and 400 nm in z under the settings described herein ⁵²." (Material and Methods section, page 18, first paragraph).

Comment 3: Further, temporal resolution is a term that has been used in the introduction and abstract, but acquisition of images was done every two hours – a time interval sufficient for several biological events to complete. This is not a major problem in conventional fluorescence microscopes, including confocal or widefield microscopy. They also need to cite the paper where lattice light sheet and structure illumination microscopy was used to study plasmodium invasion of erythrocytes (Geoghegan, Niall D., et al. "4D analysis of malaria parasite invasion offers insights into erythrocyte membrane remodeling and parasitophorous vacuole formation." *Nature communications* 12.1 (2021).

Once more, we apologized for not citing the seminal work by Prof. Torok and Prof. Rogers. Their contributions are now acknowledged in the introduction of the manuscript (see reference 13 and 34 as well as our reply to comment 1).

The reviewer correctly notes that other microscopic techniques, such as fluorescence confocal or wild-field microscopy, can capture images at 2 h time intervals during the 48 h intraerythrocytic development cycle of the parasite. We cite such studies in the manuscript (references 6 to 10). The key advantage of the approach, as outlined in the manuscript, is that it allows us to image the same individual cells throughout the developmental cycle, enabling us to capture real-time

dynamics rather than relying on pseudotemporal alignments of different cells imaged at separate time points during the experiment. Furthermore, we present a neural network based tool for the automated segmentation of infected erythrocytes, which adds an additional level to the study.

Comment 4: In Figure 4G, with the number of independent experiments being only 3, it is not advisable to carry out statistical tests. It should be at least $n=5$ (Curtis, Michael J., et al. "Experimental design and analysis and their reporting II: Updated and simplified guidance for authors and peer reviewers." *British journal of pharmacology* 175.7 (2018). Although, it can be included for the purpose of proving the point in an experimental observation.

We agree with the comment of the reviewer. The purpose of Figure 4G is to show that the B4 mutant parasite line and the parental line FCR3 exhibit a comparable capability to cytoadhere to chondroitin-4-sulfate (CSA). We have revised the figure legend to clarify this point.

"Each data point represents the average of five replicates, obtained by analyzing different sections of the CSA-coated plastic dish. The adhesion efficiency was comparable between FCR3 and B4." (page 30, end of page).

Comment 5: In Figure 7, mean number of KAHRP clusters seems to be increasing and mean fluorescence of each KAHRP cluster is also increasing. Did you check the KAHRP cluster size? Explain why this was not included. This can be easily obtained using Imaris. They should also compare this feature with the previous observation where they have seen necklace bead-like patterns for KAHRP. The spatial distribution features of KAHRP would be a useful addition in the paper.

The reviewer's observation is correct. As described in the manuscript, the number of KAHRP clusters and mean fluorescence intensity of each KAHRP cluster increase during intraerythrocytic development. It is tempting to speculate that the size of each KAHRP cluster also increases over time. In fact, previous work using atomic force microscopy demonstrated that the knob diameter increases with time in FCR3 parasite line, whereas it remains constant in other parasite lines (reference 36 in the manuscript). Given that our work focuses on FCR3 and an FCR3-derived genetically altered parasite line, we assume that the increase in KAHRP cluster fluorescence intensity over time reflects a corresponding increase in KAHRP cluster size. However, due to the size of a knob (50 to 100 nm in diameter) and the resolution of our microscopic set-up (140 nm in x and y direction), we were unable to directly investigate changes in knob size during parasite development.

Comment 6: Hence, my other concerns still stand. They need to look at the comments from the first round and address them accordingly. I am not copy pasting it here. The only novelty in the paper lies in the fact that it uses 3D-DIC images at every time point for segmentation and identification of parasites, however, whether this technique is superior to other methods reported previously is still a point of contention.

Unfortunately, we never received the full report from the first review cycle. The only comment from this reviewer that was forwarded to us pertained to the performance of the ring stage model. We hope that our response and the subsequent revisions to the manuscript addressed this concern to the reviewer's satisfaction.

Regarding the reviewer's current comment, we acknowledge their statement that the primary novelty of our work lies in the use of 3D-DIC images at every time point for segmentation and identification of parasites. However, we respectfully disagree with the implication that our approach lacks significance or advancement over previous methods. In the manuscript, we have

emphasized the advantages of our technique, particularly the ability to image the same individual cells over time and capture real-time dynamics, while maintaining cell viability. As mentioned above, this result was achieved by using DIC imaging, a technique that requires less light exposure, thereby reducing cell damage caused by light-induced phototoxicity. Additionally, we developed a neural network to segment infected erythrocytes captured. This tool solved several inherent challenges associated with analyzing DIC images, such as limited contrast and the absence of quantitative intensity information. In summary, our approach eliminates the reliance on pseudotemporal alignments, a limitation inherent in many existing methods.

Responses to reviewer's comments on manuscript COMMSBIO-24-3814B

Reviewer #1 (Remarks to the Author):

We are relieved and sincerely grateful to the reviewer that, despite a disruption in communication that was beyond anyone's control, we were able to reconnect and find an amicable and fair solution to address the reviewer's initial concerns. We appreciate the opportunity to engage with the reviewer in constructive dialogue.

Comment 1: I am quite happy with the new changes in the manuscript. The novelty lies in the label-free detection of various stages of malarial parasites using DIC images. However, caution should be exercised while comparing their paper with others, particularly with the one by Geoghegan, Niall D., et al. (Nature communications, 2021) which uses lattice light sheet, a technique that uses low dose of light. Hence, emphasis should be on the label-free DIC technique rather than on the phototoxicity of the light in the introduction and discussion. This is important because this paper has recorded DIC and fluorescence images simultaneously, for example while studying KAHRP-mEOS dynamics. This contradicts the persistent claims of negligible phototoxicity with this approach.

We accept the reviewer's perspective and have accordingly adjusted the manuscript to de-emphasize or remove claims regarding the advantages of our approach in terms of reduced phototoxicity. Instead, we now focus on the benefits of our label-free DIC technique (see introduction on page 4).

Comment 2: Also, the rebuttal has not addressed the necklace-like pattern seen for KAHRP in previous studies. It is advisable to address this in the discussion.

We respectfully note that we addressed this concern in the previous revision of the manuscript by explaining the differences in subcellular KAHRP accumulation as being due to differences in kahrp gene copy number and an overload of the protein export pathway in transiently transfected parasites with high numbers of the trans kahrp gene. For clarity, we present the relevant paragraph from the discussion below:

"Previous studies on trafficking of KAHRP observed accumulation of GFP-tagged KAHRP fusion proteins in the parasite ER, the digestive vacuole and as a necklace-like pattern in the parasitophorous vacuolar lumen 44. In contrast, we did not observe fluorescence clusters in compartments other than around the host cell plasma membrane. We attribute this discrepancy to differences in kahrp gene copy number and an overload of the protein export pathway in transiently transfected parasites with high numbers of the trans kahrp gene." (page 16, second paragraph).

Comment 3. Furthermore, the rebuttal states the XY resolution as 160 nm in comment 2, then goes on to state it as 140 nm in comment 5.

We sincerely apologize for the typographical error in the rebuttal letter. The correct XY resolution is 160 nm, as accurately stated in the manuscript (Page 18, end of the first paragraph).

Nonetheless, the generalisation remains valid, especially in their novel use of the DIC-based label-free approach in identifying parasites.

Thank you for your encouraging words. We appreciate the reviewer's recognition of the novelty and significance of our approach.